# *Citrus reticulata* Olive Oil: Production and Nutraceutical Effects on the Cardiovascular System in an In Vivo Rat Model of Metabolic Disorder

**DOI:** 10.3390/nu16183172

**Published:** 2024-09-19

**Authors:** Jacopo Spezzini, Valerio Ciccone, Monica Macaluso, Ylenia Pieracci, Guido Flamini, Sandra Donnini, Vincenzo Calderone, Lara Testai, Angela Zinnai

**Affiliations:** 1Department of Pharmacy, University of Pisa, Via Bonanno Pisano 6, 56126 Pisa, Italy; jacopo.spezzini@phd.unipi.it (J.S.); ylenia.pieracci@phd.unipi.it (Y.P.); guido.flamini@unipi.it (G.F.); vincenzo.calderone@unipi.it (V.C.); lara.testai@unipi.it (L.T.); 2Department of Life Science, University of Siena, Via Aldo Moro 2, 53100 Siena, Italy; valerio.ciccone@unisi.it (V.C.); sandra.donnini@unisi.it (S.D.); 3Department of Agriculture, Food and Environment (DAFE), University of Pisa, Via del Borghetto 80, 56124 Pisa, Italy; angela.zinnai@unipi.it; 4Interdepartmental Research Center “Nutraceuticals and Food for Health”, University of Pisa, Via del Borghetto 80, 56124 Pisa, Italy

**Keywords:** *Citrus reticulata*, extra virgin olive oil, cardiovascular risk, by-products, bioeconomy

## Abstract

Recently, there has been significant exploration into the utilization of food by-products as natural reservoirs of bioactive substances, particularly in the creation of functional foods naturally enriched with antioxidants. *Citrus* peels represent a viable option for formulating enhanced olive oils that contribute to a healthier diet, due to their bioactive compound content. This study aimed to (i) ascertain the compositional characteristics of *Citrus reticulata* olive oil (CrOO) and (ii) assess its nutraceutical properties in rats with metabolic disorder induced by 3 weeks of feeding with a high-fat diet (HFD). The results showed a peculiar phytochemical composition, thanks to the contribution of citrus peels, which are excellent bio-products. In addition, it demonstrated HFD-induced weight gain (18 ± 2% for HFD vs. 13 ± 0.9% for CrOO) and showed protective effects on fasting blood glucose levels (90.2 ± 3.8 mg/dL for HFD vs. 72.3 ± 2.6 for CrOO). Furthermore, a reduction in cardiovascular risk (total cholesterol/HDL cholesterol = 5.0 ± 0.3 for HFD vs. 3.8 ± 0.3 for CrOO) and an improvement in myocardial tissue function were observed, as well as a significant reduction in inflammatory mediators such as iNOS, COX-2, and mPGES-1 in aortic vessel tissues, thus preserving endothelial function at the vascular level.

## 1. Introduction

*Citrus* by-products are valuable sources of essential bioactive compounds with potential applications in animal feed, processed foods, and healthcare [1].

Citrus fruits are a rich source of bioactive secondary metabolites, which are primarily produced by plants as a defense strategy against insects and microbial pathogens [2,3,4]. Their phytochemical content imparts noteworthy health protection and disease prevention properties. *Citrus* fruits are rich in flavonoids, carotenoids, and limonoids, with approximately 95% of flavonoids being flavanones, such as naringenin, hesperetin, and eriodictyol [5,6,7].

Epidemiological, clinical, and preclinical evidence highlights the nutraceutical benefits of *Citrus* fruits on the cardiovascular system, demonstrating vasorelaxing and cardioprotective effects [8]. Naringenin and hesperetin have also shown SIRT1-mediated anti-aging properties in nematode and yeast models [6,9]. Studies suggest that citrus fruits contribute to cardiovascular health by improving the cardiometabolic profile, reducing plasma levels of total and low-density lipoprotein (LDL) cholesterol and triglycerides, and limiting the increase in weight caused by the high-fat diet [10].

Moreover, citrus fruits are rich in carotenoids, offering at least 110 different carotenes and xanthophylls. These pigments play a crucial role in photosynthesis and help prevent photooxidation. Carotenoids provide numerous health benefits, reducing the risk of diseases such as tumors and heart conditions and ophthalmological issues [11,12].

Limonoids represent another relevant class of biologically active molecules in *Citrus*, and exhibit anticancer effects and possess pharmacological properties like antimicrobial, antioxidant, antidiabetic, and insecticidal effects [13,14]. Given the growing interest in bioactive compounds in foods, there is a considerable focus on formulating combined nutraceutical compounds in healthy functional foods to prevent and contrast chronic diseases [15].

Fortified food (FF) emerges as an effective tool to enhance nutritional intake, as it is widely accessible, socially acceptable, quickly introduced, and generally perceived as safe and cost-effective [16,17]. In this context, innovatively produced fortified olive oils were obtained using an innovative method, incorporating cryomacerated citrus peels or leaves (*C. aurantium* or *C. lemon*) during the olive oil extraction process [10]. This approach resulted in oils enriched with the nutraceutical properties typical of citrus fruits, enhancing the organoleptic profile in terms of smell complexity and hedonic response [10]. Based on the previous evaluation regarding the addition of citrus by-products during olive oil extraction, an innovative product was developed, using *Citrus reticulata* peels, according to the previously described system [10]. The present research aimed to determine the chemical profile of *Citrus reticulata* olive oil (CrOO) and evaluate its nutraceutical qualities in rats with metabolic disorders associated with a high-fat diet.

## 2. Materials and Methods

### 2.1. Plant Material

*Citrus reticulata* fruits were harvested at full maturity during the 2020/2021 crop season, following the protocol described by Ascrizzi et al. [18]. Olives (Moraiolo cultivar) were collected during the 2021/2022 crop season and supplied by an organic farm in Tuscany (Azienda Agricola Val Di Lama, Pontedera (PI), Italy), and characterized according to the method outlined by Macaluso et al. [19].

### 2.2. Citrus Peel Cryomaceration and Citrus Olive Oil Extraction 

*Citrus* fruits were prepared and cryomacerated following the methods described in previous studies [10]. After cryomaceration, the citrus peels were directly added to the olives at a ratio of 25% by weight before milling to produce Citrus-enriched Olive Oil (CrOO). The extraction of olive oil was carried out using a micro oil mill (Spremioliva mod. C30, Mori-TEM srl, Barberino Tavarnelle (FI), Italy) according to the extraction protocol detailed in a previous publication [19].

### 2.3. Citrus Olive Oil Chemical Analyses

#### 2.3.1. Chemical Quality Standards

Free acidity, peroxide value, and spectrophotometric indices were determined following the official analytical methods detailed in the European legal standards [20].

#### 2.3.2. Total Phenols 

Total phenols (TP) were extracted using a liquid–liquid extraction method with a methanol solution (80:20 *v*/*v*), following the procedure outlined in a previous study [19].

#### 2.3.3. Free-Radical Scavenging Capacity (FRSC)

The free-radical scavenging capacity (FRSC) was assessed using two methods: the DPPH (2,2-diphenyl-1-picrylhydrazyl) assay, referred to as FRSCDPPH [10], and the ABTS (2,2’-azino-bis(3-ethylbenzothiazoline-6-sulfonic acid)) assay (FRSCABTS). The radical solutions were prepared according to the procedure previously described [10].

#### 2.3.4. Intensity of Bitterness (IB) 

The intensity of bitterness was assessed using the method of Gutiérrez Rosales et al. [21], without modifications. 

#### 2.3.5. Carotenoids and Chlorophylls

Total carotenoids and chlorophylls were measured according to the procedure outlined by Minguez Mosquera, without any modifications [22]. 

#### 2.3.6. Analysis of Hydroxytyrosol and Tyrosol

The analysis of hydroxytyrosol and tyrosol was conducted using reverse-phase high-performance liquid chromatography (RP-HPLC) under the chromatographic conditions previously reported [23]. 

#### 2.3.7. Extraction and Detection of Tocopherols (Vitamin E)

As reported [10], tocopherols were extracted in the dark. Specifically, α-, β-, γ-, and δ-tocopherol isoforms were analyzed by isocratic reverse-phase high-performance liquid chromatography (RP-HPLC) using a Shimadzu LC-20AD apparatus (Shimadzu Europa GmbH, Duisburg, Germany) with an electrochemical detector (Metrohm model 791, Varese, Italy) and a glassy carbon electrode. 

#### 2.3.8. Headspace-Solid Phase Microextraction Analysis

The volatile emissions of both the control and the *C. reticulata* olive oil were analyzed by headspace-solid phase microextraction (HS-SPME) following the procedure previously described [18]. 

#### 2.3.9. Gas Chromatography–Mass Spectrometry Analyses

The gas chromatography–electron ionization mass spectrometry (GC-EIMS) analysis was carried out using an Agilent 7890B gas chromatograph (Agilent Technologies Inc., Santa Clara, CA, USA), equipped with an Agilent HP-5MS capillary column (30 m × 0.25 mm, 0.25 µm film thickness) and an Agilent 5977B single quadrupole mass detector. The analytical conditions included an oven temperature program that increased from 60 °C to 240 °C at a rate of 3 °C/min, with the injector and transfer line temperatures set at 250 °C and 240 °C, respectively. Helium was used as the carrier gas, at a flow rate of 1 mL/min. Mass spectra were acquired in full scan mode over a range of 30–300 *m*/*z* with a scan time of 1.0 s. Peak identification was performed by comparing retention times with those of reference compounds and evaluating linear retention indices relative to a series of n-alkanes (C8–C27). Further identification was supported by matching spectra with commercial mass spectral libraries (NIST 14 and ADAMS 2007) and a custom library of pure substances, essential oil components, and relevant literature data [24].

### 2.4. In Vivo Evaluation of the Nutraceutical Properties of Enriched Extra Virgin Olive Oil

In vivo experimentation was performed according to European (EEC Directive 2010/63) and Italian (D.L. 4 March 2014 n.26) legislation. The project protocol was discussed at the Animal Ethics Committee of the University of Pisa and received approval from the Committee of the Italian Ministry of Health (number protocol 144/2019-PR, 18 February 2019). Moreover, ARRIVE guidelines were followed. The purpose of ARRIVE guidelines is to improve the quality and reproducibility of in vivo research [25]. Animals were caged with free access to food and water and with a 12 h dark/light cycle. The experiments were carried out on adult male Wistar rats (10 weeks old, ENVIGO, Nanakramguda Village, Hyderabad, India) weighing between 305 and 360 g; each treatment group was arranged in order to present the same average weight at the beginning of the treatment. Experiments were performed only on male rats in order to avoid hormonal interferences. According to the literature [26], animals were divided into 4 groups (n = 5/group) and treated for 21 days. The first group received a standard diet (STD, ENVIGO; for composition see Table 1; this pellet was polyphenol-free), the second group received a high-fat diet (HFD, U8220P version 0151, SAFE; composition reported in Table 1; ingredients: casein, lard, sucrose, maltodextrin, pregelatinized cornstarch, crude cellulose, soybean oil, potassium citrate, dicalcium phosphate, pre-mixture of minerals PM AIN 93M/G 3.5%, pre-mixture of vitamins PV AIN 93M/G 1%, calcium carbonate, L-cysteine, choline bitartrate); and the third and fourth groups received, respectively, HFD + extra virgin olive oil (CEOO) 2% p/p and HFD + CrOO 2% p/p. The amount of CEOO was calculated based on the regulatory authorities’ guidelines regarding the daily intake of polyphenols [27]. All 4 groups ate the same amount and the same calories, and water intake, food intake, and body weight were evaluated three times a week for each animal, contemporaneously with the administration of supplementations. At the end of the treatment, after 21 days, animals were starved overnight and at the end of the starvation period they were anesthetized with isoflurane and a blood glucose test was performed using blood from the caudal vein; subsequently, animals were sacrificed by cervical dislocation. EDTA tubes (BD Vacutainer, Franklin Lakes, NJ, USA) were used to collect intracardiac blood, and organs (heart, epididymal white adipose tissue, and intrascapular brown adipose tissue and liver) were recovered, weighed, and stored for further analysis. The lipid panel (high-density lipoproteins (HDL) and low-density lipoproteins (LDL), cholesterol and triglycerides) was evaluated rapidly using intracardiac blood with a Cobas b101 instrument (Roche Diagnostics, Rotkreuz, Switzerland). The remaining blood was centrifugated at 3200 RPM for 10 min in order to obtain plasma that was stored for further investigations.

#### 2.4.1. Functional Analysis of Cardiac Mitochondrial Membrane Potential

Harvested hearts were finely minced into (2–3 mm^3^) pieces in isolation buffer (composition: Sucrose 250 mM, Tris 5 mM, EGTA 1 mM; pH 7.4); the whole procedure was carried out in an ice bath. Heart pieces then received three cycles of homogenization of about 20 s each. To obtain functional mitochondria, the homogenates were then centrifuged at 1090× *g* for 3 min at 4 °C, the pellets were removed and discarded, and the supernatants were further centrifuged at 11,970× *g* for 10 min at 4 °C. The pellet was preserved and resuspended in ice-cold isolation buffer without EGTA and centrifuged again at the same conditions. The mitochondrial fraction was represented by a pellet that was immediately resuspended in 400 μL of buffer and transferred in a tube. The amount of protein in the mitochondrial fraction was assessed following the Bradford assay (Bio-Rad, Hercules, CA, USA) with a microplate reader (EnSpire, PerkinElmer, Waltham, MA, USA). 

To assess the mitochondrial membrane potential (ΔΨm) of the isolated mitochondria, a potentiometric method was employed. Tetraphenylphosphonium (TPP+), a lipophilic cation, was detected using a TPP+-sensitive mini-electrode (WPI, TipTPP, Sarasota, FL, USA) in conjunction with a reference electrode (WPI, Sarasota, FL, USA) and software for data collection (Biopac Inc., Goleta, CA, USA). Mitochondria (1 mg protein/mL) were rapidly added to an incubation buffer with the following composition: KCl 120 mM, K_2_HPO_4_ 5 mM, Hepes 10 mM, succinic acid 10 mM, MgCl_2_ 2 mM, TPP+Cl− 10 μM, and pH 7.4. The mixture was maintained in suspension with continuous magnetic stirring. The membrane potential value was calculated according to the following experimental equation, derived from the Nernst Equation (1), and following the protocol previously reported by Flori et al. (see reference [10]):(1)$${\Delta \Psi }_{m}=60\times \log\frac{\frac{V_{0}{\left[{TPP}^{+}\right]}_{0}}{{\left[{TPP}^{+}\right]}_{t}}-V_{t}-K_{0}P}{V_{m}P+K_{i}P}$$

The mitochondrial membrane potential was measured for all the animals in the different groups, and data were expressed as mean ± SEM.

#### 2.4.2. Western Blot

Protein analysis was performed on tissue samples following established protocols [10], with all experiments repeated at least three times. Densitometric analysis of the immunoblots was performed using NIH Image J 1.48v software. The data, presented as arbitrary density units (ADU) ± standard deviation (SD), were normalized against β-actin, which served as the loading control [28]. 

#### 2.4.3. Immunohistochemistry

Immunohistochemical analysis was performed using seven μm thick cryostat sections obtained from tissue samples. The tissue sections were initially fixed in 4% paraformaldehyde (PFA) for 20 min, followed by treatment with 3% H_2_O_2_ for 10 min. Afterwards, they were rinsed three times in PBS for 5 min each and blocked for non-specific binding using 5% goat serum. The sections were then incubated at 4 °C for 18 h with a rabbit monoclonal anti-CD68 antibody and diluted 1:100 in PBS containing 0.05% goat serum. Following three additional 5 min washes in PBS with 0.05% goat serum, the sections were treated with biotinylated goat anti-rabbit IgG for 60 min. After washing again (three times for 5 min each in PBS with 0.05% goat serum), the sections were exposed to streptavidin-conjugated HRP for 10 min. To visualize the antibody binding, 3,3-diaminobenzidine tetrahydrochloride (DAB) was applied for 8 min, resulting in a brown color. The sections were then counterstained with hematoxylin and mounted using Eukitt^®^ Quick-hardening mounting medium (Merck KGaA, Darmstadt, Germany) [10].

#### 2.4.4. Hematoxylin and Eosin Staining

The sections (7 μm thick) were stained with hematoxylin for 3 min and then washed under running water for 5 min. Subsequently, eosin was added for 6 min, followed by a 5 min wash with running water. This was followed by dehydration in increasingly graded alcohols up to xylene. Finally, the sections were mounted using Eukitt^®^ Quick-hardening mounting medium. 

### 2.5. Statistical Analysis 

The data are presented as the mean values from three independent experiments. Statistical differences between group means were analyzed using one-way ANOVA (CoStat, Version 6.451, CoHort Software, Pacific Grove, CA, USA). Post hoc comparisons were conducted using either Tukey’s or Bonferroni’s test, with a significance threshold of *p* < 0.05. UV and MS data analyses were conducted using Xcalibur 3.1 software.

## 3. Results and Discussion

### 3.1. Oil Chemical Characterization 

The characterization of Citrus-enriched Olive Oil (CrOO), produced by adding mandarin peels to olives during extraction, was performed in comparison to the control Cold-Extracted Virgin Olive Oil (CEVOO). This evaluation considered not only the quality criteria specified in trade standards but also the phytochemical characteristics, including the bioactive compounds detected in the CrOO.

#### 3.1.1. Legal Quality Parameters

The quality parameters outlined in the trade standard are designed to classify oils into various categories. As indicated in Table 2, despite the olives being relatively ripe, with a maturity index of 3.8 on a scale of 7.0, the olive oil samples met the legal quality criteria. Key parameters, such as free acidity, peroxide value, and spectrophotometric indices (K232, K270, and ΔK), were all below the maximum limits established by EU regulations for EVOO.

Table 3 presents analytical results for the different types of olive oils, specifically CEVOO and CrOO. The significance level indicates the statistical significance of the differences observed.

CEVOO and CrOO both exhibit significantly lower free acidity (0.44), meeting the specified limit. The differences among the three types were not statistically significant.

Both CEVOO and CrOO demonstrate low peroxide levels (8.90 and 8.80, respectively), well below the specified limit. The differences among the three types were not statistically significant. CEVOO (1.98) fell within the specified limit, while CrOO (2.10) was slightly higher, but both CEVOO (0.14) and CrOO (0.18) were still within the specified limit. 

#### 3.1.2. Phytochemical Composition

Chlorophylls and carotenoids play a crucial role in determining the color of olive oil, a quality factor that can significantly impact consumer preferences. Additionally, the levels of these pigments are associated with factors such as olive variety, processing methods, and storage conditions. Beyond their influence on visual appeal, both chlorophylls and carotenoids are noteworthy for their potential positive effects on human health. The obtained results indicated significant variations in the total carotenoid content between the two oils, as outlined in Table 4. This divergence in carotenoid levels suggested distinct compositions or concentrations of these pigments in the analyzed oils. The differences observed may be attributed to factors such as the specific olive varieties used, variations in processing techniques, or differences in storage conditions affecting the stability of these pigments. Understanding and monitoring the levels of chlorophylls and carotenoids not only contributes to the assessment of olive oil quality but also provide insights into the potential health benefits associated with these bioactive compounds. The observed variations underscore the need for further investigation into the specific factors influencing pigment content, as this knowledge can guide producers in optimizing processing methods and storage conditions to maintain both visual appeal and potential health-promoting properties in olive oil. The choice of qualitative markers can also help to standardize production by selecting an optimal harvest period in order to obtain an ideal quantity of bioactive compounds which will then characterize this type of product.

As expected, CrOO displayed the highest carotenoid content among the studied oils. Moving on to the broader category of oil pigments and total chlorophyll, significant differences were noted, with CEVOO exhibiting the highest concentration of chlorophylls. The health-promoting attributes of olive oil stem from its unique composition, encompassing a well-balanced ratio of unsaturated fatty acids alongside minor components such as phenolic compounds and tocopherols. These components not only contribute to the nutritional quality of the product but also influence its shelf life.

The tocopherol profile and phenolic content of the Citrus-enriched Olive Oil (CrOO) were assessed. Under the experimental conditions, the addition of cryomacerated citrus peels did not significantly alter the oil samples regarding total tocopherols or their isoforms (alpha, gamma, and delta). Similarly, the total phenol content showed no substantial differences among the samples, with values around 135 ppm of gallic acid. However, CrOO exhibited lower concentrations of hydroxytyrosol and tyrosol compared to Cold-Extracted Virgin Olive Oil (CEVOO). These observations were supported by the free-radical scavenging capacity (FRSC) results, which indicated that citrus olive oils had lower antioxidant activity than the control in both analytical methods. Additionally, the study evaluated compounds related to olive oil bitterness, using the Intensity of Bitterness (IB) as a parameter. 

#### 3.1.3. Volatile Composition

The complete composition of the volatile emission of CrOO, reported in Table 5, consisted of 11 compounds, representing 100% of the aroma profile, almost all belonging to the class of monoterpene hydrocarbons. Within this class, limonene was undoubtedly the chief component, accounting for 90.4% of the headspace composition; however, non-negligible amounts of myrcene, α-pinene, and sabinene were also revealed, even though they did not exceed the 5%. The occurrence of those major chemical compounds in the volatile emission of the CrOO was most likely determined by mandarin peels [29]. Conversely, the presence of non-terpene derivative, albeit found in very low percentages, was probably attributed to the olive oil. In particular, (E)-2-hexenal, accounting for 0.3% in the *C. reticulata* product, represented the major component detected in the volatile emissions of the extra virgin olive oil produced using the same olive fruits (CEVOO) and the same technological process employed to obtain the mandarin olive oil.

### 3.2. Effects of CEVOO and CrOO Supplementation on the Cardiometabolic Profile of HFD-Fed Rats

According to the literature [10,30,31], the high-fat diet determined a significant increase in body weight gain (18 ± 2%) compared to the standard chow (10 ± 1%). Animals that received supplementation with CEVOO showed a reduced gain in body weight (12 ± 0.6%). Likewise, at the end of the protocol, animals fed with HFD + CrOO showed a significant containment of body weight gain (13 ± 0.9%), superimposable with that observed in animals fed with STD and with the CEVOO supplementation (Figure 1).

A high-fat dietary regimen is a widely used and recognized experimental model to induce significant metabolic alterations in rodents. In fact, together with the increase in body weight, this diet determines a metabolic dysfunction, characterized by altered glycemic and lipidic homeostasis, as well as cellular modifications prodromic to increased cardiovascular risk.

Indeed, animals fed with HFD for three weeks showed, at the end of the period, higher levels of blood glucose and cholesterol, triglycerides, and an increase in cardiovascular risk value (Total Cholesterol/HDL-cholesterol) compared with STD, as reported in Table 6.

Of note, blood glucose and triglycerides levels were markedly reduced in animals that received CEVOO supplementation compared to the HFD group; conversely, the levels of total cholesterol, LDL, HDL, and non-HDL-cholesterol were not modified. However, the cardiovascular risk was markedly reduced. The CrOO-supplementation presented a superimposable profile to the group fed with HFD + CEOO; in particular, the blood glycemia and triglycerides were significantly reduced, and the cardiovascular risk was also markedly contained compared to the HFD group (Table 6). 

Consistently, LDL levels presented a reducing trend, while HDL values were slightly increased (Table 6). The liver showed the presence of numerous fat deposits, reflecting a significant increase in liver weight. On the other hand, treatment with oils did not significantly alter this parameter. For other organs, such as the heart, HFD did not caused any significant change in weight.

HFD induced a significant increase in liver weight; nevertheless, CEVOO supplementation was not able to reduce this parameter. HFD also determined a significant increase in the weight of white adipose tissue and a significative reduction in brown adipose tissue weight, and the supplementation with CEVOO showed a trend, even if not significative, in improving these parameters, while CrOO ameliorated only the weight of the brown adipose tissue (Figure 2). 

Lastly, mitochondrial isolation was performed to assess mitochondrial membrane potential. This value is an important indicator of the health of heart mitochondria and their ability to produce ATP despite the stress induced with HFD. In a healthy animal, physiological values of ΔΨ were −191 ± 3.9 mV and HFD determined a depolarization, increasing ΔΨ at −178.5 ± 3.8 mV. The supplementation with CEVOO and CrOO showed a trend of restoring the mitochondrial membrane potential (−185 ± 4 and −186 ± 3 mV, respectively) (Figure 3).

On the basis of these results and in accordance with the literature, the HFD regimen effectively induced profound metabolic alterations, determining an increase in total cholesterol, triglycerides, HDL and LDL, and blood glucose levels, typical of the metabolic disorders [10,19].

Animals that received CEVOO or CrOO supplementation presented a significant reduction in blood glucose levels, presenting values like the ones observed with the standard diet, demonstrating the hypoglycemic and protective properties of these supplementations. The whole lipid profile (total cholesterol, triglycerides, HDL and LDL-cholesterol) was significantly increased with the administration of the HFD, while supplementation with either CEVOO or CrOO significantly reduced the cardiovascular risk. Regarding triglyceride levels, the administration of the CEVOO was also able to restore levels similar to the ones observed with STD, while CrOO induced a reduction in terms of triglyceride levels. Regarding the evaluation of the health of the heart mitochondria, both CEVOO and CrOO partially rescued mitochondrial membrane potential, leading to the conclusion that this transforming strategy does not impair the well-known nutraceutical profile of the extra virgin olive oil. 

### 3.3. Effects of CEVOO and CrOO Supplementation on Levels of Pro-Inflammatory Markers in Rat Aortic Vessels

Systemic and local inflammation play pivotal roles in the progression of cardiovascular disease [32], and the high expression of inflammatory mediators is a risk factor for arterial disease [33]. In rat aortic tissues, we assessed the expression of COX-2 and mPGES-1, inducible enzymes involved in the arachidonic acid cascade, leading to the production of PGE2, as well as inducible nitric oxide synthase (iNOS). Our results demonstrated that an HFD significantly increased the protein expression of COX-2 and mPGES-1 compared to an STD (Figure 4A,C,D). This upregulation highlights the pro-inflammatory effects of an HFD, which could contribute to the development and progression of cardiovascular disease. Interestingly, supplementation with CrOO was effective in reducing the inflammatory tone, as evidenced by the decreased expression of both COX-2 and mPGES-1 (Figure 4A,C,D). This suggests that CrOO has potent anti-inflammatory properties that can counteract the harmful effects of an HFD. Conversely, CEVOO alone did not reduce the HFD-induced overexpression of mPGES-1. Additionally, CrOO proved to be as effective as CEVOO alone in reducing the overexpression of iNOS induced by HFD (Figure 4B). Anti-inflammatory and antioxidant effects of virgin olive oil on vascular tissue were mainly associated with the levels of hydroxytyrosol and tyrosol [34,35,36]. Although CrOO showed lower concentrations of hydroxytyrosol and tyrosol than CEVOO, both olive oils were shown to reduce vascular inflammation, and CrOO did so with a larger effect, suggesting that its efficacy could be attributed to the vitamin content of carotenoids [37] and tocopherols [38,39], widely reported as potential antioxidants and anti-inflammatory agents.

Further morphological assessment of the aortic tissues through hematoxylin and eosin staining revealed no significant variations in tissue morphology across the different diet and supplementation groups (Figure 5A). This duration is not sufficient to induce morphological alterations in the aortic structure, as widely reported in the literature, where some studies extend the treatment to 8–12 weeks [40,41]. This suggests that the observed biochemical changes did not translate into gross morphological alterations within the timeframe of our study. Subsequent CD56 staining was performed to evaluate immune cell infiltration in the aortic vascular wall. Our analysis showed that CD56 expression was absent in the STD, CEVOO, and CrOO groups, while positive CD56 staining was observed in the HFD group (Figure 5B). This indicates that an HFD promotes immune cell infiltration into the aortic wall, a process that is mitigated by CrOO supplementation. The absence of CD56 staining in the CEVOO and CrOO groups further supports the anti-inflammatory effects of these supplements.

Altogether, our study highlights the detrimental impact of a high-fat diet on inflammation within the aortic tissues, as evidenced by the upregulation of COX-2, mPGES-1, and iNOS, and increased immune cell infiltration. CrOO supplementation emerged as a potent anti-inflammatory intervention, effectively reducing the expression of inflammatory markers and preventing immune cell infiltration. While CEVOO and CrOO showed some efficacy in modulating specific inflammatory pathways, CrOO demonstrated a broader protective effect. These findings suggest that dietary interventions, particularly with CrOO, could be a valuable strategy in mitigating inflammation and potentially reducing the risk of cardiovascular disease associated with high-fat diets. 

## 4. Conclusions

Our findings revealed that, despite the high maturity index, the olive oil samples used in this study met the stringent quality standards for extra virgin olive oil (EVOO). Both CEVOO and CrOO exhibited significantly low free acidity and peroxide levels, with no significant differences among the oil types. The high carotenoid content in CrOO and high chlorophyll concentration in CEVOO were notable, influencing both visual appeal and potential health benefits. Despite these differences, both oils demonstrated robust profiles in maintaining the quality and health-promoting attributes of olive oil, including their balanced fatty acid composition and the presence of minor compounds like phenolics and tocopherols. While *Citrus* cryomacerated peels did not significantly alter tocopherol profiles or total phenolic content, CrOO had lower hydroxytyrosol and tyrosol concentrations compared to CEVOO, correlating with its lower free-radical scavenging capacity. Furthermore, CrOO showed high levels of limonene, a volatile compound peculiar to citrus fruits and endowed with renowned beneficial effects, that was almost absent in the CEVOO.

Despite the short duration of the treatment (3 weeks), which could—at least in part—lead to underestimation of the real nutraceutical impact of CEVOO as well as CrOO on obesogenic dietary regimen, in in vivo studies, CEVOO showed potential in reducing body weight gain, blood glucose, and triglycerides, while CrOO showed similar effects, especially in containing cardiovascular risk. The HFD significantly impacted metabolic parameters, increasing total cholesterol, triglycerides, HDL, LDL, and blood glucose levels. Supplementation with CEVOO or CrOO mitigated these effects, highlighting their protective properties. Both oils helped to restore mitochondrial membrane potential, suggesting maintained nutraceutical profiles. Additionally, CrOO reduced inflammatory markers like COX-2, mPGES-1, and iNOS in aortic tissues, further supporting its anti-inflammatory efficacy.

In summary, our study underscores the negative impact of a high-fat diet on cardiovascular health and inflammation, confirming the benefits of CEVOO supplementation and highlighting that the technological strategy implemented to exploit citrus by-products does not compromise the well-known nutraceutical value on the cardiovascular system of extra virgin olive oil, suggesting, rather, its potential as a dietary intervention to combat the negative effects of high-fat diets.

## Figures and Tables

**Figure 1 nutrients-16-03172-f001:**
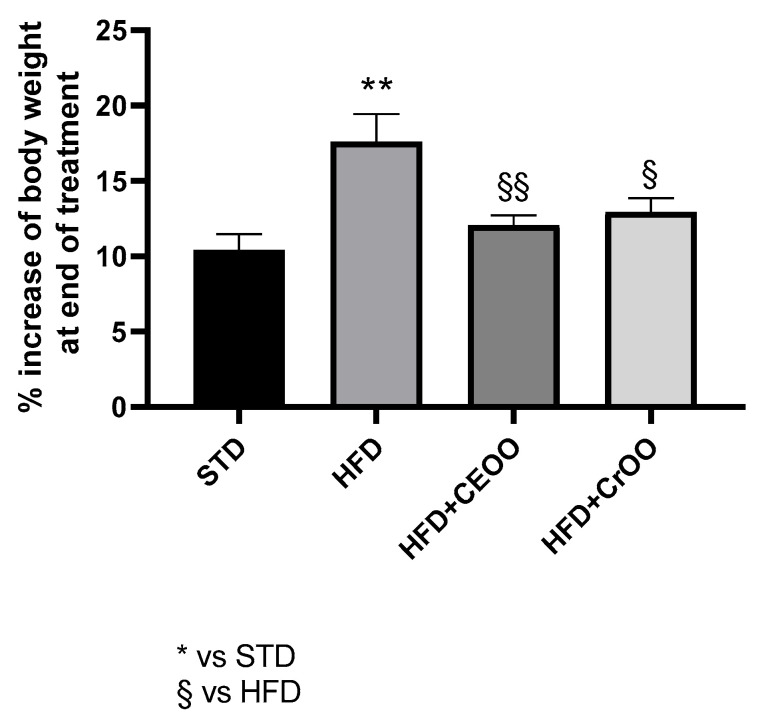
Time course weight increase (%) during the treatment period (n = 5). * indicates a statistically significant difference between the HFD group and the STD group. § indicates statistically significant difference vs. HFD. Single symbol corresponds to *p* ≤ 0.05. Double symbol corresponds to *p* ≤ 0.01.

**Figure 2 nutrients-16-03172-f002:**
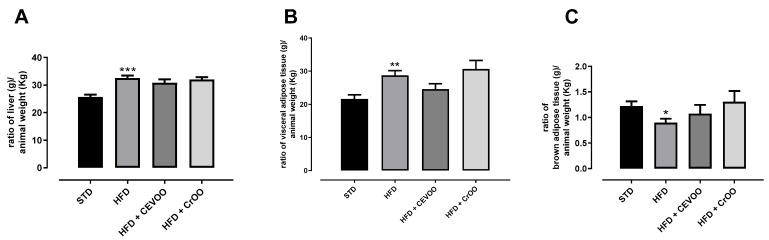
(**A**) Ratio of liver weight (g) to animal weight (kg); (**B**) ratio of epididymal white adipose tissue weight (g) to animal weight (kg); and (**C**) ratio of intrascapular brown adipose tissue weight (g) to animal weight (kg) at the end of the treatments. * Indicates a statistically significant difference between the HFD group and the STD group. * Corresponds to *p* < 0.05, ** corresponds to *p* < 0.01 and *** *p* < 0.001 (n = 5).

**Figure 3 nutrients-16-03172-f003:**
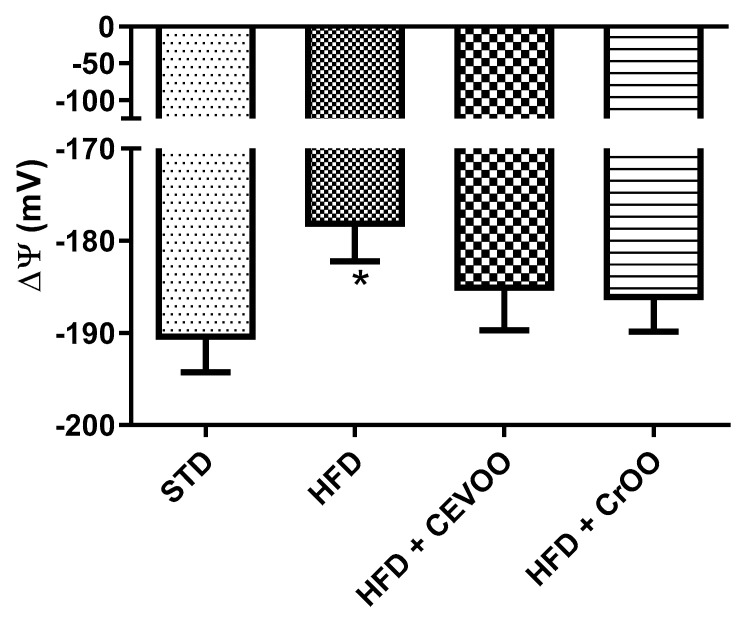
Changes in mitochondrial membrane potential between the different treatment groups (n = 5). * Indicates a statistically significant difference between the HFD group and the STD group. * Corresponds to *p* < 0.05.

**Figure 4 nutrients-16-03172-f004:**
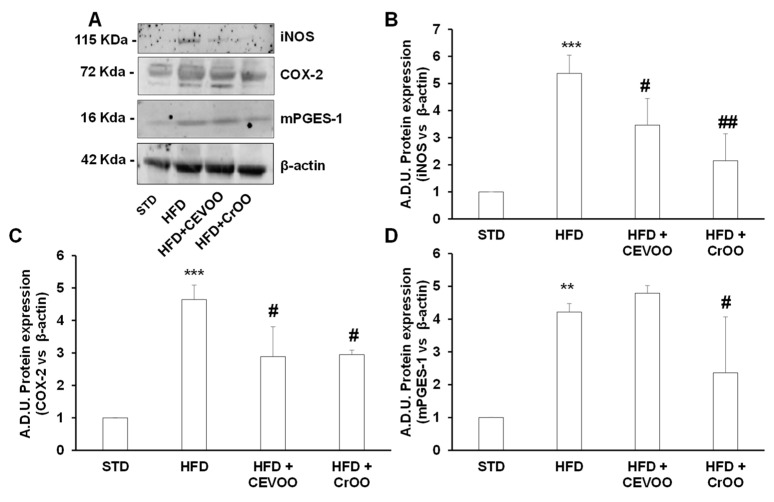
Activity of CEVOO and CrOO supplementation on inflammatory markers in aortic tissues of rats fed a high-fat diet. (**A**) Representative blots of inflammatory factors (iNOS, COX-2, and mPGES-1) in aortic vessel tissues. Each lane was loaded with 50 μg of total protein. (**B**) Quantification of iNOS. (**C**) Quantification of COX-2. (**D**) Quantification of mPGES-1. Data (ratio of A.D.U. normalized on β-actin) are reported as fold change vs. STD, assigned a value of 1 (n = 3/group). ** *p* < 0.01 and *** *p* < 0.001 vs. STD; # *p* < 0.05 and ## *p* < 0.01 vs. HFD. Comparisons among means in (**B**–**D**) were performed with the Bonferroni’s post hoc test.

**Figure 5 nutrients-16-03172-f005:**
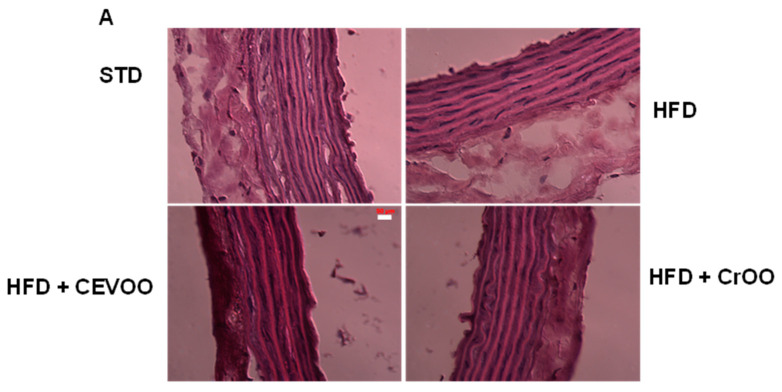
Morphology and CEVOO and CrOO supplementation activity on aortic vascular wall inflammation from rats fed with/without HFD. Magnification ×40. Scale bar 50 µm for all panels. (**A**) Morphology of aortic tissue under hematoxylin and eosin staining. (**B**) Evaluation of aortic sections by immunohistochemistry. Arrow in HFD panel shows CD56 staining. Representative section of three rats/experimental group.

**Table 1 nutrients-16-03172-t001:** Macro- and micro-nutrient composition of standard diet (STD) and high-fat diet (HFD).

	STD	HFD
Protein (%)	14.3	12.9
Fat (%)	4.0	19.2
Carbohydrate (%)	48.0	50.2
Calories from Protein (%)	20.0	12.1
Calories from Fat (%)	13.0	40.7
Calories from Carbohydrate (%)	67.0	47.2
Na (mg/kg)	1000.0	2234.1
K (mg/kg)	6000.0	5284.3
Mg (mg/kg)	2000.0	1294.0
Ca (mg/kg)	7000.0	6312.5
Mn (mg/kg)	100.0	47.7
Fe (mg/kg)	175.0	253.2
Cu (mg/kg)	15.0	18.5
Zn (mg/kg)	70.0	48.7
P (mg/kg)	6000.0	5144.2
Cl (mg/kg)	3000.0	3508.1
Vitamin A (IU/g)	6.0	7.4
Vitamin E (IU/kg)	120.0	29.8
Vitamin D_3_ (IU/g)	0.6	1.0
Vitamin K_3_ (mg/kg)	20.0	13.3
Vitamin B_1_ (mg/kg)	12.0	4.2
Cholesterol (mg/kg)	-	12488

**Table 2 nutrients-16-03172-t002:** Olive fruit characterization. Data are expressed as mean ± confidence interval (n = 3) at *p* = 0.05.

Parameter	
**Maturity index**	3.8 ± 0.1
**Dry Matter (%)**	53.66 ± 0.08
**Oil Content (% d.m.)**	17.70 ± 0.03

**Table 3 nutrients-16-03172-t003:** Chemical characterization of CEVOO, CrOO, and EVOO legal limit regulations.

	EVOO	CEVOO	CrOO	*Significance Level* ^1^
**Free acidity (% oleic acid *w*/*w*)**	≤0.80	0.44 ^a^	0.44 ^a^	n.s.
**Peroxide index (mEq. O_2_/kg oil)**	≤20.00	8.90 ^a^	8.80 ^a^	n.s.
**K232**	≤2.50	1.98 ^b^	2.10 ^a^	**
**K270**	≤0.22	0.14 ^b^	0.18 ^a^	**
**ΔK**	≤0.01	0.00	0.00	n.s.

^1^ Significance level—** *p* < 0.01; n.s.: not significant (*p* ≥ 0.05). Within the same row, parameters sharing the same letter do not have a significantly different mean value.

**Table 4 nutrients-16-03172-t004:** Phytochemical characterization of control (CEVOO) and *Citrus reticulata* olive oil (CrOO).

	CEVOO	CrOO	*Significance Level* ^1^
**Total carotenoids (ppm lutein)**	4.44 ^b^	9.34 ^a^	***
**Total chlorophylls (ppm pheophytin)**	9.34 ^a^	5.23 ^b^	***
**α-tocopherol (ppmVitamin E)**	110	133	**
**γ-tocopherol (ppmVitamin E)**	2.7	3.2	**
**δ-tocopherol (ppmVitamin E)**	0.76	1.4	**
**Total phenols (ppm gallic acid)**	133	138	n.s
**FRSC ABTS (µmol TEAC/mL)**	0.30 ^a^	0.34 ^a^	n.s
**FRSC DPPH (µmol TEAC/mL)**	0.24 ^a^	0.22 ^a^	n.s
**Hydroxtyrosol (ppm)**	0.06 ^a^	0.02 ^b^	**
**Tyrosol (ppm)**	2.4 ^a^	1.3 ^b^	**
**Intensity of bitterness**	0.55 ^a^	0.57 ^a^	n.s

^1^ Significance level—** *p* < 0.01 *** *p* < 0.001; n.s: not significant (*p* ≥ 0.05). Within the same row, parameters sharing the same letter do not have a significantly different mean value.

**Table 5 nutrients-16-03172-t005:** Volatile composition of CrOO and CEVOO.

Compounds	l.r.i. ^1^	Class	Relative Abundance (%) ± SD	
			CEVOO	CrOO
hexanal	802	nt	2.7 ± 0.4	- ^2^
*p*-xylene	870	nt	1.5 ± 0.01	-
(*E*)-2-hexenal	892	nt	82.7 ± 2.32	0.3 ± 0.0
*o*-xylene	897	nt	1.6 ± 0.2	-
3-ethyl-1,5-octadiene (isomer 1)	898	nt	0.7 ± 0.0	-
3-ethyl-1,5-octadiene (isomer 2)	901	nt	0.5 ± 0.0	-
α-pinene	933	mh	-	1.9 ± 0.0
1-ethyl-4-methylbenzene	965	nt	0.3 ± 0.4	
sabinene	973	mh	-	1.4 ± 0.0
β-pinene	977	mh	-	0.1 ± 0.0
myrcene	991	mh	-	3.8 ± 0.0
octanal	1003	nt	-	0.8 ± 0.0
α-phellandrene	1005	mh	-	0.2 ± 0.0
δ-3-carene	1011	mh	-	0.9 ± 0.0
1,2,4-trimethylbenzene	1025	nt	0.2 ± 0.2	-
limonene	1029	mh	1.0 ± 0.7	90.4 ± 0.0
(*E*)-β-ocimene	1052	mh	2.0 ± 0.3	
terpinolene	1089	mh	-	0.1 ± 0.0
*n*-nonanal	1102	nt	0.5 ± 0.8	-
(*E*)-4,8-dimethylnona-1,3,7-triene	1116	nt	0.8 ± 0.1	-
(*E*)-2-dodecene	1205	nt	0.4 ± 0.5	-
decanal	1206	nt	-	0.1 ± 0.0
cyclosativene	1368	sh	0.2 ± 0.3	-
α-copaene	1376	sh	2.3 ± 0.2	-
valencene	1492	sh	1.3 ± 0.0	-
(*E*,*E*)-α-farnesene	1507	sh	0.4 ± 0.5	-
liguloxide	1532	os	0.7 ± 0.1	-
Monoterpene hydrocarbons (mh)			3.0 ± 0.4	98.8 ± 0.1
Sesquiterpene hydrocarbons (sh)			4.2 ± 0.5	-
Oxygenated sesquiterpenes (os)			0.7 ± 0.1	-
Other non-terpene derivatives (nt)			92.1 ± 0.1	1.2 ± 0.1
Total identified (%)			99.9 ± 0.1	100.0 ± 0.0

^1^ Linear retention index experimentally determined on an HP-5MS capillary column; ^2^ not detected.

**Table 6 nutrients-16-03172-t006:** Summary of data on lipid panel and glycemic profile at the end of treatment.

	STD	HFD	HFD + CEVOO	HFD + CrOO
Total cholesterol (mmol/L)	1.98 ± 0.09	3.00 ± 0.19 ***	2.74 ± 0.17	2.96 ± 0.25
Triglycerides (mg/dL)	71.4 ± 4.1	107.8 ± 14.2 *	67.7 ± 2.9 ^§^	86.0 ± 6.7
HDL-cholesterol (mg/dL)	49.3 ± 3.2	28.9 ± 1.8 ***	30.2 ± 1.8	31.4 ± 3.1
LDL-cholesterol (mg/dL)	34.4 ± 7.1	71.9 ± 4.5 ***	68.1 ± 6.1	62.4 ± 4.5
non-HDL-cholesterol (mg/dL)	29.1 ± 1.8	81.1 ± 5.2 ***	82.2 ± 5.8	83.2 ± 9.5
Cardiovascular risk (Total Cholesterol/HDL-cholesterol)	1.6 ± 0.1	5.0 ± 0.3 ***	3.7 ± 0.2 ^§§^	3.8 ± 0.3 ^§^
Fasting blood glucose (mmol/L)	3.66 ± 0.26	5.02 ± 0.21 ***	3.79 ± 0.24 ^§§^	4.01 ± 0.15 ^§^

* Indicates a statistically significant difference compared to the STD group. ^§^ Indicates a statistically significant difference compared to the HFD group. Single symbol corresponds to *p* < 0.05, double symbol to *p* < 0.01, and triple symbol to *p* < 0.001.

## Data Availability

The original contributions presented in the study are included in the article, further inquiries can be directed to the corresponding author.

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
