# Peer review of "Citrus reticulata Olive Oil: Production and Nutraceutical Effects on the Cardiovascular System in an In Vivo Rat Model of Metabolic Disorder"

_nutrients, 2024, doi:10.3390/nu16183172_

Round 1
Reviewer 1 Report
Comments and Suggestions for Authors
The manuscript firstly investigated the composition and quality parameters of an Extra Virgin olive oil enriched with Citrus reticulate peels CrOO, compared to the Extra Virgin olive oil CEVOO obtained from the same olives and with the standard parameters for EVOO. Carotenoids, chlorophylls and total phenolics were quantified by a spectrophotometric method. Tocopherols were quantified by HPLC and volatiles by headspace solid phase microextraction and GC-MS, while other quality parameters where assessed by standard methods, and antioxidant activity by well-known methods. Generally, this part was adequately addressed and the results indicates the specific profile of an olive oil enriched with phytochemicals from citrus peel (please see the specific comments below). The second part of the manuscript presents the in vivo experiment, aiming to demonstrate the cardio protective effect of CrOO and CEVOO, on male rats feed with a high fat diet for 21 days. Various relevant parameters were determined and the results are convincing regarding the body weight and tissue weight, as well as the relevant blood parameters related to lipid metabolism.
The scientific hypothesis is correct, the experimental design is adequate and the manuscript is well-written and appropriate to the topic of the journal. Some questions, recommendations, and suggestions are listed below:
1. The abstract should be more informative (include more results)
2. Line 49 – only a small number of carotenoids possess provitamin A activity, please correct. I could not access reference 8. There are so many very good, comprehensive and recent reviews on carotenoids. I can recommend one or two (Bohn, T.; Bonet, M.L.; Borel, P.; Keijer, J.; Landrier, J.-F.; Milisav I., … Dulińska-Litewka, J. Mechanistic aspects of carotenoid health benefits –where are we now? Nutr Res Rev. 2021, 34(2), 276-302. doi: 10.1017/S0954422421000147; Böhm, V.; Lietz, G.; Olmedilla-Alonso, B.; Phelan, D.; Reboul, E.; Bánati, D.; .... Bohn, T. From carotenoid intake to carotenoid blood and tissue concentrations – implications for dietary intake recommendations, Nutrition Reviews 2021, 79(5):544-573. doi: 10.1093/nutrit/nuaa008)
3. Line 127 – 25-75 ng on column? Please be more specific, usually the calibration curve is given in terms of concentration. The equation of the standard curves and the correlation coefficient should be provided.
4. Did the authors perform a quantification also for volatile compounds or only the relative abundance (%)? I know that this is a difficult task.
5. Line 312 – Please reconsider this phrase. Olive oil is known as a predominantly monounsaturated oil.
6. Lines 331-334 – The number of volatile compounds is really low. Citrus reticulate have hundreds of volatile compounds. Please discuss the limitations of this study regarding the determination of the volatiles.
7. What method was used for the determination of tyrosol and hydroxytyrosol? I could not find this information on the manuscript.
8. One can expect a deeper characterization of CrOO regarding phenolics compounds and carotenoids considering that HPLC system was available in the laboratory and these to classes of compounds were considered by the authors to be associated with the cardio protective effect of CrOO. Moreover, the authors did a very similar work on other citru enriched OO in which this part was more consistent.
9. In my opinion authors should make a statistic comparison also between CEVOO and CrOO (Fig. 2, 3, 4) in order to emphasize the superiority of CrOO, because the effects of CEVOO are already well documented
10. The standard iNOS antibody does not have any signal. Is the low signal for HFD and HFD+CEVOO lysates characteristic of the iNOS antibody at the expected molecular weight range? Moreover, it seems that there was a problem with the protocol. The blocking reagent used clumped together, and the antibodies bound to it or the gel or reagents were contaminated with bacteria (dark dots or positive signals on the Western blot).
11. Lines 451-460. Probably the duration of the experiment is not long enough to produce macroscopic morphological changes. Please discuss
12. I recommend to be consistent with the notations, two abbreviations are used along the manuscript CrOO and CrVOO
13. Authors should present and discuss all the methodological limitations of this study
14. Some references are not correctly cited, please revise (e.g. 7, 8)
Author Response
Reviewer 1
The manuscript firstly investigated the composition and quality parameters of an Extra Virgin olive oil enriched with Citrus reticulate peels CrOO, compared to the Extra Virgin olive oil CEVOO obtained from the same olives and with the standard parameters for EVOO. Carotenoids, chlorophylls and total phenolics were quantified by a spectrophotometric method. Tocopherols were quantified by HPLC and volatiles by headspace solid phase microextraction and GC-MS, while other quality parameters where assessed by standard methods, and antioxidant activity by well-known methods. Generally, this part was adequately addressed and the results indicates the specific profile of an olive oil enriched with phytochemicals from citrus peel (please see the specific comments below). The second part of the manuscript presents the in vivo experiment, aiming to demonstrate the cardio protective effect of CrOO and CEVOO, on male rats feed with a high fat diet for 21 days. Various relevant parameters were determined and the results are convincing regarding the body weight and tissue weight, as well as the relevant blood parameters related to lipid metabolism.
The scientific hypothesis is correct, the experimental design is adequate and the manuscript is well-written and appropriate to the topic of the journal. Some questions, recommendations, and suggestions are listed below:
- The abstract should be more informative (include more results)
ANSWER: We thank the referee for the suggestion. We have added more results in order to improve the clarity.
- Line 49 – only a small number of carotenoids possess provitamin A activity, please correct. I could not access reference 8. There are so many very good, comprehensive and recent reviews on carotenoids. I can recommend one or two (Bohn, T.; Bonet, M.L.; Borel, P.; Keijer, J.; Landrier, J.-F.; Milisav I., … Dulińska-Litewka, J. Mechanistic aspects of carotenoid health benefits –where are we now? Nutr Res Rev.2021, 34(2), 276-302. doi: 10.1017/S0954422421000147; Böhm, V.; Lietz, G.; Olmedilla-Alonso, B.; Phelan, D.; Reboul, E.; Bánati, D.; .... Bohn, T. From carotenoid intake to carotenoid blood and tissue concentrations – implications for dietary intake recommendations, Nutrition Reviews 2021, 79(5):544-573. doi: 10.1093/nutrit/nuaa008).
ANSWER: We thank the referee for the suggestion. We have modified the text according to the referee's observation.
- Line 127 – 25-75 ng on column? Please be more specific, usually the calibration curve is given in terms of concentration. The equation of the standard curves and the correlation coefficient should be provided.
ANSWER: We thank the referee for the suggestion. We have added the requested information.
- Did the authors perform a quantification also for volatile compounds or only the relative abundance (%)? I know that this is a difficult task.
ANSWER: The volatile composition was expressed as relative abundance (%) since the absolute quantification particularly for HS-SPME analyses, is very challenging and is some cases not possible due to the unavailability of pure compounds and difficulties in recreating a matrix identical to the one analyzed to be used for calibration
- Line 312 – Please reconsider this phrase. Olive oil is known as a predominantly monounsaturated oil.
ANSWER: We rewrote the sentence according to the referee's suggestion.
- Lines 331-334 – The number of volatile compounds is really low. Citrus reticulate have hundreds of volatile compounds. Please discuss the limitations of this study regarding the determination of the volatiles.
ANSWER: the limited number of volatile compounds detected in C. reticulata olive oil is due to the great prevalence of limonene that represented almost 90% of the whole volatile emission. Indeed, the volatile composition was expressed as relative abundance (%), thus the strong prevalence of a particular compound undoubtedly determines the reduction of the relative amount of minor components. Moreover C. reticulata olive oil and C. reticulata essential oil are produced with completely different production methods that could explain the presence of a limited number of compounds in the obtained oil, if compared to the pure essential oil.
- What method was used for the determination of tyrosol and hydroxytyrosol? I could not find this information on the manuscript.
ANSWER: The referee is right; it was a mistake. We added the paragraph 2.3.6
- One can expect a deeper characterization of CrOO regarding phenolics compounds and carotenoids considering that HPLC system was available in the laboratory and these to classes of compounds were considered by the authors to be associated with the cardio protective effect of CrOO. Moreover, the authors did a very similar work on other citru enriched OO in which this part was more consistent.
ANSWER: The referee is right and we thank him for the suggestion. Unfortunately, we had problems with the instrumental apparatus and were unable to carry out a more in-depth characterization at the moment. However, we will try to do it in the future.
- In my opinion authors should make a statistic comparison also between CEVOO and CrOO (Fig. 2, 3, 4) in order to emphasize the superiority of CrOO, because the effects of CEVOO are already well documented
ANSWER: We thank the reviewer for the suggestion, but we performed a statistical comparison between the two extra virgin olive oils without observing statistically significant differences. From our prospective the technological strategy implemented to exploit a citrus by-product without compromising the well-known nutraceutical value on cardiovascular system of extra virgin olive oil.
- The standard iNOS antibody does not have any signal. Is the low signal for HFD and HFD+CEVOO lysates characteristic of the iNOS antibody at the expected molecular weight range? Moreover, it seems that there was a problem with the protocol. The blocking reagent used clumped together, and the antibodies bound to it or the gel or reagents were contaminated with bacteria (dark dots or positive signals on the Western blot).
ANSWER: As reported in the literature, the expression of iNOS under control conditions is very low in both cardiac tissues (DOI: 10.1186/1758-5996-3-37; doi: 10.1097/ALN.0 b013e31818d7e5a; doi: 10.1006/jmcc.1999 .1005) and in the aorta, where it is found in trace amounts in some layers (adventitia) and absent in others (media) (doi.org/10.1152/ajpheart.2000.279.6.H2743). Furthermore, iNOS expression increases as the animal ages (10.1097/ALN.0b013e31818d7e5a), so it is not surprising that iNOS is not detected in controls. In addition, the samples were processed simultaneously following the same protocol, which reduces the probability of degradation of the protein of interest, and because exposure to HFD increases iNOS expression and co-treatment with oils significantly reduces it, we are confident in the proper execution and reliability of the experiment.
About potential contamination of the blocking agent, this cannot be ruled out with certainty, but the reagents for the Western blot are prepared fresh for each experiment.
- Lines 451-460. Probably the duration of the experiment is not long enough to produce macroscopic morphological changes. Please discuss
ANSWER: We fully agree with the reviewer's comment. In our model, only certain parameters of metabolic syndrome are replicated. Undoubtedly, to observe morphological changes in the rat aorta, it would be necessary to extend the duration of the high-fat diet. We have addressed this aspect by adding the following (lines 451-453):
“This duration is not sufficient to induce morphological alterations in the aortic structure, as widely reported in the literature, where some studies extend the treatment to 8-12 weeks (DOI: 10.1016/j.exger.2021.111543; doi: 10.1002/phy2.268). However, the finding that a monocytic infiltrate is already observed after three weeks only in aortas isolated from HFD-treated rats and not in those isolated from rats treated with HFD and the oils strengthens the experimental model and supports the anti-inflammatory activity of the oils.”
- I recommend to be consistent with the notations, two abbreviations are used along the manuscript CrOO and CrVOO
ANSWER: The referee is right; it was a mistake.
- Authors should present and discuss all the methodological limitations of this study
ANSWER: We added the methodological limitations in the “Conclusions” paragraph. Despite the short duration of the treatment (3 weeks), that could -at least in part - underestimate the real nutraceutical impact of CEVOO as well as CrOO on obesogenic dietary regimen; in in vivo studies…
- Some references are not correctly cited, please revise (e.g. 7, 8)
ANSWER: We have modified following the referee's suggestion

Reviewer 2 Report
Comments and Suggestions for Authors
Interesting paper comparing extra virgin olive oil and mandarin olive oil! I have to say that the methodology is fine but the no. of samples is very limited.
The authors claim enhancement of sensory attributes but I have not seen any sensory analysis by a panel group. We need to have a better image of the maturity of the citrus fruits, the weight of the fruits when they were harvested and besides the acidity we need to know the sugars and the sugars to acidity ratio. The same for extra virgin olive oil.
No quantities have been given in 2.1, 2.2. Very limited no. of samples and not clear of the quantities.
The authors state that peels stayed in contact with dry ice and this is CO2s?
Please explain.
The authors state an in vivo evaluation please change to toxicological evaluation and please state if it is acute.
Any reference for cold extracted olive oil?
Olive oil extraction was carried out by a pilot olive oil mill. Please state the conditions and also state the conditions for the cold extracted olive oil extracted along with mandarin olive oil.
Comments on the Quality of English LanguageVery good
Author Response
Reviewer 2
Comments and Suggestions for Authors:
Interesting paper comparing extra virgin olive oil and mandarin olive oil! I have to say that the methodology is fine but the no. of samples is very limited.
The authors claim enhancement of sensory attributes but I have not seen any sensory analysis by a panel group. We need to have a better image of the maturity of the citrus fruits, the weight of the fruits when they were harvested and besides the acidity we need to know the sugars and the sugars to acidity ratio. The same for extra virgin olive oil.
ANSWER: Thanks for the suggestion. Unfortunately we did not have enough sample to perform a panel test, but we used the analysis of volatile compounds to investigate the differences and or similarities actually present in the two samples. The characterization of the fruits used was carried out but we do not consider it necessary for the purposes of this work since the attention is focused on the product obtained and on the comparison with EVOO (we can consider adding these data in the supplementary material).
No quantities have been given in 2.1, 2.2. Very limited no. of samples and not clear of the quantities.
The authors state that peels stayed in contact with dry ice and this is CO2s?
Please explain.
ANSWER: Reference reports in more detail the extraction system used, the quantities used and the processes. To avoid self-plagiarism, we have chosen not to detail the entire procedure.
The authors state an in vivo evaluation please change to toxicological evaluation and please state if it is acute.
ANSWER: We thank the reviewer for the comment, but the aim of this study was to evaluate the cardiometabolic protective properties after a chronic administration (3 weeks) of either CrVOO or CEVOO in an in vivo model of obesity using rats fed with high fat diet.
Any reference for cold extracted olive oil?
ANSWER: Reference 11 reports in detail the extraction system used.
Olive oil extraction was carried out by a pilot olive oil mill. Please state the conditions and also state the conditions for the cold extracted olive oil extracted along with mandarin olive oil.
ANSWER: Again, see Reference 11.
Comments on the Quality of English Language
Very good
Submission Date
25 July 2024
Date of this review
06 Aug 2024 17:40:53

Reviewer 3 Report
Comments and Suggestions for Authors
The manuscript “Citrus reticulata Olive Oil: Production and Nutraceutical Properties on the Cardiovascular System” is well written and discussed.
However, some aspects should be improved:
- Why the authors do not compare the Citrus reticulata olive oil with Citrus reticulata essential oil (composition and biological potential)?
- In the abstract, the authors reported that “The research demonstrated that the production of Citrus reticulata olive oil exhibited protective effects concerning glucose and serum lipid levels, adipocyte metabolic activity, myocardial tissue functionality, oxidative stress markers, and endothelial function at the vascular level” Considering the antioxidant capacity, it would no longer make sense to work with another type of bioactive compounds, such as phenolic compounds?
- 2.3.7-2.3.8, In respect to GC/MS analysis, please include more information. How do the authors know that these conditions are best, or possibly better, for this type of compounds? And also the fibre. Why 10 minutes?
- In the GC/MS technique, what are the pure samples used? In respect to the quantification of volatiles, the quantification was performed in full scan or by ions? Please include information about the expression of the results, for example, how obtained the relative abundance (%)? In this article, the authors reported a low number of volatiles, why they were not quantified?
- Please include a chromatogram of GC/MS with the peak identification.
- 2.4. In the in vivo evaluation…, please include the name and reference of the Animal Ethics Committee acceptation.
- In the discussion section, please insert more references to compare/support the results herein obtained.
For these reasons, I consider the manuscript accept with major revisions on Nutrients.
Author Response
Reviewer 3
Comments and Suggestions for Authors
The manuscript “Citrus reticulata Olive Oil: Production and Nutraceutical Properties on the Cardiovascular System” is well written and discussed.
However, some aspects should be improved:
- Why the authors do not compare the Citrus reticulata olive oil with Citrus reticulata essential oil (composition and biological potential)?
Answer: We would like to thanks the referee for the question. We preferred to compare Citrus reticulata olive oil with the volatile emission of Citrus reticulata fruits since both analyses were performed with the same method (HS-SPME). Moreover, the C. reticulata olive oil and C. reticulata essential oil are produced with completely different production methods that make the comparison of those products not possible. The different protocols could also explain the presence of a reduced number of compounds detected in the obtained oil, if compared to the pure essential oil.
- In the abstract, the authors reported that “The research demonstrated that the production of Citrus reticulata olive oil exhibited protective effects concerning glucose and serum lipid levels, adipocyte metabolic activity, myocardial tissue functionality, oxidative stress markers, and endothelial function at the vascular level” Considering the antioxidant capacity, it would no longer make sense to work with another type of bioactive compounds, such as phenolic compounds?
Answer: The phenolic compounds introduced by the citrus enrichment are likely to be significant contributors to the health benefits observed in this study. Phenolic compounds are well-known for their antioxidant and anti-inflammatory properties, which can play a crucial role in promoting health. However, it is important to recognize that the oleic acid component of the extra virgin olive oil is also vital, as it is associated with various health benefits, including cardiovascular protection and anti-inflammatory effects.
Given this dual importance—both the phenolic and oleic components contribute to the oil’s overall health effects—the study aimed to evaluate the enriched and original oils in their entirety rather than isolating individual components. This holistic comparison allows for a more accurate assessment of how the combined effects of all components interact to produce the health benefits reported. It aligns with the study's broader objective of presenting a food product that not only preserves but potentially enhances the health properties of the original olive oil. This approach provides a comprehensive understanding of how the enriched oil might offer superior or complementary health advantages compared to its unenhanced counterpart.
- 2.3.7-2.3.8, In respect to GC/MS analysis, please include more information. How do the authors know that these conditions are best, or possibly better, for this type of compounds? And also the fibre. Why 10 minutes?
Answer: The employed GC-MS conditions were selected since they allowed the best separation of the analytes, obtaining chromatographic peaks with a good resolution. Concerning the sampling, the type of fibre was selected because of the nature of the sample, while the sampling time was experimentally determined to obtain an optimal adsorption of the volatile compounds, avoiding both under- and over-saturation of the fibre and of the mass spectrometer.
- In the GC/MS technique, what are the pure samples used? In respect to the quantification of volatiles, the quantification was performed in full scan or by ions? Please include information about the expression of the results, for example, how obtained the relative abundance (%)? In this article, the authors reported a low number of volatiles, why they were not quantified?
Answer: As reported in the material and method section, the analysis of the volatile compounds was performed in full scan mode and the peak identification was based on a comparison of the retention times with those of pure samples (limonene, myrcene, α-pinene, sabinene, β-pinene, α-phellandrene, cyclosativene δ-3-carene, (E)-β-ocimene, terpinolene, α-copaene, valencene, nonanal, decanal, octanal, hexanal, (E)-2-hexenal) comparing their linear retention indices relative to the series of n-hydrocarbons (C8-C27). Moreover, a computer matching against commercial (NIST 14 and ADAMS 2007) and laboratory-developed mass spectra libraries built up from pure substances and components of commercial essential oils of known composition and MS literature data was also performed The composition of the components detected in the oil samples was expressed as the mean of the relative abundance (%) of 3 replicates. Unfortunately, the absolute quantification, particularly for HS-SPME analyses, is very challenging and is some cases not possible due to the unavailability of pure compounds and difficulties in recreating a matrix identical to the one analyzed to be used for calibration.
- Please include a chromatogram of GC/MS with the peak identification.
Answer: We thank the referee for the suggestion but we believe that the chromatograms add nothing more than the data reported in the table, so if the presence of the chromatograms is very important for the referee we can include it in the supplementary section.
- 2.4. In the in vivo evaluation…, please include the name and reference of the Animal Ethics Committee acceptation.
Answer: We discussed the project protocol at the Animal Ethics Committee of the University of Pisa and we received the approval by the Committee of Italian Ministry of Health. In the lines 153-155 we reported the sentence “In vivo experimentation was performed according to European (EEC Directive 2010/63) and Italian (D.L. 4 March 2014 n.26) legislation (number protocol 144/2019-PR, 18/02/2019)” referring to the number of protocol of the project and the date of approval.
- In the discussion section, please insert more references to compare/support the results herein obtained.
Answer: According to the reviewer’s comment, we added new references in the discussion section.
For these reasons, I consider the manuscript accept with major revisions on Nutrients.
Submission Date
25 July 2024
Date of this review
02 Aug 2024 17:01:57

Round 2
Reviewer 1 Report
Comments and Suggestions for Authors
The authors responded to all the questions and requested corrections. I am satisfied with the response and the modifications.
Author Response
We thank the reviewer for his comments!
Reviewer 2 Report
Comments and Suggestions for Authors
Many thanks for the authors to have replied but not specifically in some cases.
Regarding the sensory analysis of a trained panel the authors claimed that not enough sampling was there. The authors cannot quote sesnory results without mentioning a sensory analysis by a trained sensory panel.
They have been asked regarding extraction methodology and referred to their review paper which mentions the extraction technology under 5.1.2. This is their research paper and are supposed to quote the conditions of the pilot plant extraction technology.
If these 2 issues are not addressed next time paper will be rejected and I strongly believe it is a very good paper worth of publishing.
Comments on the Quality of English Language
Minor English corrections
Author Response
Reviewer 2
Many thanks for the authors to have replied but not specifically in some cases.
Regarding the sensory analysis of a trained panel the authors claimed that not enough sampling was there. The authors cannot quote sensory results without mentioning a sensory analysis by a trained sensory panel.
ANSWER: We thank the referee for the question. We have followed the recommendation and removed the references to sensory characteristics, which although highlighted with other methodologies, were not subject to panel testing. In the future we will certainly carry out this analysis as well.
They have been asked regarding extraction methodology and referred to their review paper which mentions the extraction technology under 5.1.2. This is their research paper and are supposed to quote the conditions of the pilot plant extraction technology.
ANSWER: We thank the referee for the observation. There was an error in the citations and we have corrected it. For the olive oil extraction system, we refer to our previous articles (doi:10.3390/nu12061557; doi:10.3390/molecules24010065). We hope we have cleared up the misunderstanding.
If these 2 issues are not addressed next time paper will be rejected and I strongly believe it is a very good paper worth of publishing.

Reviewer 3 Report
Comments and Suggestions for Authors
The manuscript “Citrus reticulata Olive Oil: Production and Nutraceutical Properties on the Cardiovascular System” is well written and discussed and the corrections were performed as requested.
For these reasons, I consider the manuscript accept as is on Nutrients.
Author Response
We thank the referee.